# ADAPTIVE SHARPNESS-AWARE PRUNING FOR ROBUST SPARSE NETWORKS

**Anna Bair** [*]
Carnegie Mellon University
abair@cmu.edu

**Hongxu Yin, Maying Shen, Pavlo Molchanov, Jose Alvarez**
NVIDIA
{dannyy, mshen, pmolchanov, josea}@nvidia.com

## ABSTRACT

Robustness and compactness are two essential attributes of deep learning models that are deployed in the real world. The goals of robustness and compactness may seem to be at odds, since robustness requires generalization across domains, while the process of compression exploits specificity in one domain. We introduce *Adaptive Sharpness-Aware Pruning (AdaSAP)*, which unifies these goals through the lens of network sharpness. The AdaSAP method produces sparse networks that are robust to input variations which are *unseen at training time*. We achieve this by strategically incorporating weight perturbations in order to optimize the loss landscape. This allows the model to be both primed for pruning and regularized for improved robustness. AdaSAP improves the robust accuracy of pruned models on image classification by up to +6% on ImageNet C and +4% on ImageNet V2, and on object detection by +4% on a corrupted Pascal VOC dataset, over a wide range of compression ratios, pruning criteria, and network architectures, outperforming recent pruning art by large margins.

## 1 INTRODUCTION

Deep neural networks have increasingly been used in many applications such as autonomous driving. Unlike the controlled environments in which these models are trained, test-time inference presents new challenges including noisy real-world data and latency and memory constraints. These challenges have led to recent efforts in network robustness and compression, but through largely separate lines of work.

Robustness to input variation unseen during training is especially important for deployed deep learning models in safety-critical applications such as autonomous driving, where, for instance, artifacts such as dirt or snow may obscure the camera image. More generally, in computer vision applications, this input variation falls into categories including distribution shifts, image corruptions, adversarial attacks, and label noise (Hendrycks & Dietterich, 2019; Azulay & Weiss, 2018; Carlini & Wagner, 2016; 2017; Hendrycks et al., 2018; Steinhardt et al., 2017). In attempts to address these challenges, the vision community has collected and released datasets to assess model robustness (Hendrycks & Dietterich, 2019; Recht et al., 2019) and investigated and improved performance on a variety of types of

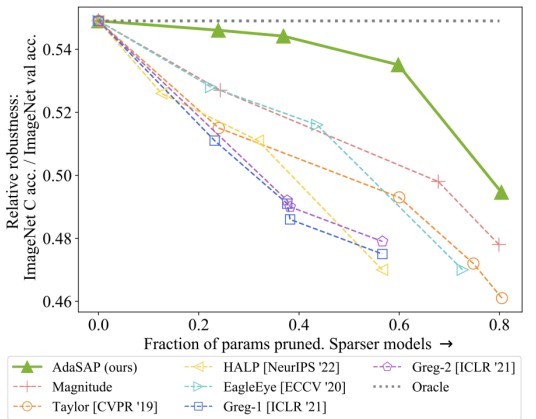

Figure 1: Robustness of pruned models trained on ImageNet-1K drastically degrades on ImageNet-C as pruning ratio increases for many SOTA pruning methods. AdaSAP reduces the degradation in robust performance relative to standard validation performance. We approach the grey dashed line, which indicates an ideal scenario in which robust performance does not degrade at higher rates than validation performance.

---

[*] Work done during an internship at NVIDIA.

Figure 2: AdaSAP is a three step process that takes as input a dense pretrained model and outputs a sparse robust model. The process can be used with any pruning method.

robustness (Xie et al., 2021; Zhou et al., 2022; Szegedy et al., 2013; Madry et al., 2017). Despite significant progress in this area, most existing robustness work focuses on dense networks (Guo et al., 2023; Zhou et al., 2022).

Running large models imposes substantial computational burdens that can be mitigated via model compression methods. Given the observation that neural networks often include redundant computation at inference, this line of work aims to reduce network inference costs through techniques such as knowledge distillation (Hinton et al., 2015; Molchanov et al., 2022; Yin et al., 2020), quantization (Gholami et al., 2021; Banner et al., 2018; Idelbayev et al., 2021; Cai et al., 2020), pruning (Han et al., 2015b; Yang et al., 2021; Yu et al., 2019; Molchanov et al., 2016; 2019; Shen et al., 2022a), and network adaptation (Molchanov et al., 2022; Dai et al., 2019; Yang et al., 2018). Here we focus on pruning due to its simple procedure for network speedup. The pruning process retains neurons deemed important over the training distribution, so it is unsurprising that pruned networks can have reduced out-of-distribution (OOD) generalization, as observed in Liebenwein et al. (2021); Hooker et al. (2019). The effort to jointly address robustness and compression remains an open problem.

The goal of our work is to produce compact and robust neural networks. In particular, we emphasize robustness to input variations that are *unseen at training time*, in contrast with methods such as robust pruning (Sehwag et al., 2020; Vemparala et al., 2021; Zhao & Wressnegger, 2022) which assume access to the input variations at training time. It might seem as if the goals of sparsity and robustness are at odds since one aims to exploit the current dataset and task for extreme compactness whereas the other strives for maximal generalization. However, our method leverages a new flatness-based optimization procedure that both primes the network for pruning and improves network robustness. Specifically, flatness can mitigate the loss in performance during the pruning procedure and regularize the network towards improved robustness.

We introduce *Adaptive Sharpness-Aware Pruning (AdaSAP)*, a three-step algorithm that prunes a network so that it is robust to OOD inputs. (1) We introduce the novel *adaptive weight perturbations* which are incorporated into the optimization procedure to adaptively penalize sharpness in order to prepare the network for pruning, (2) apply any pruning method to the model, and (3) continue training the model while uniformly penalizing sharpness across the network to encourage robustness.

AdaSAP significantly improves the relative robustness over prior pruning art, seen in Figure 1. As models are pruned (with SOTA methods) more, their performance on corrupted images suffers disproportionately as compared to their validation performance. This highlights the lack of robustness preservation in recent SOTA pruning methods and AdaSAP's success at reducing this degradation[1].

Our contributions can be summarized as follows:

- We introduce *AdaSAP*, a sharpness-aware pruning and fine-tuning process to jointly optimize for sparsity and robustness in deep neural networks.

- We propose novel *adaptive weight perturbations* that prepare a network for both pruning and robustness to OOD inputs. This strategy actively manipulates the flatness of the loss surface so as to minimize the impact of removing neurons with lower importance scores.

- We demonstrate state-of-the-art performance compared to prior art by noticeable margins across a wide range of setups, covering (i) two tasks (classification and detection), (ii) two OOD types (image corruption and distribution shift), (iii) four networks (ResNet50 and MobileNet V1/V2 for

---

[1]Relative robustness in Fig. 1 refers to the ratio between the accuracy on corrupted images and the standard validation accuracy at a given prune ratio. A dense model in this setup retains $55\%$ of its validation performance when given corrupted images (location of grey dashed line).

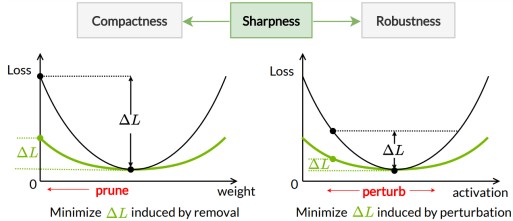

Figure 3: **(Left)** Before pruning, encourage neurons that will be pruned to lie within a flat minimum, since their removal will affect the loss less. **(Right)** After pruning, promote robustness by encouraging flatness across the network.

**Algorithm 1** AdaSAP Optimization Iteration

**Require:** model weights $\mathbf{w}$ partitioned into neurons $\mathbf{w}_i$, training batch $b$, $\rho$ bounds ($\rho_{\min}$, $\rho_{\max}$), loss $L$, score function $\psi$, learning rate $\eta$
**for** each neuron $\mathbf{w}_i$ **do**
    $s_i = \psi(\mathbf{w}_i)$          ▷ Score
    Compute $\rho_i$ as in Eq. 2    ▷ Determine perturbation ball size
    Compute $\hat{\epsilon}_i$ as in Eq. 5 ▷ Optimal perturbation
    $g_i \approx \nabla_{\mathbf{w}_i} L_{b,\mathbf{w}}(\mathbf{w}_i)|_{\mathbf{w}_i + \hat{\epsilon}_i}$    ▷ Gradient approximation
**end for**
$\mathbf{w} = \mathbf{w} - \eta \mathbf{g}$
**return** $\mathbf{w}$

classification, SSD512 for detection), (iv) two pruning paradigms (parameter-based and latency-based), and (v) sparsity ratios spanning from around 20% to 80%.

- We present a detailed evaluation of robustness and compactness and show, as a first encouraging attempt, that both goals can be unified through the lens of sharpness, with analysis to encourage the community to continue pursuing this direction.

## 2 RELATED WORKS

**Robustness.** Researchers and practitioners care about many types of robustness, including adversarial robustness, robustness to distribution shift, robustness to image corruptions such as weather effects, and robustness to label noise or data poisoning (Szegedy et al., 2013; Madry et al., 2017; Hendrycks & Dietterich, 2019; Azulay & Weiss, 2018; Carlini & Wagner, 2016; 2017; Hendrycks et al., 2018; Steinhardt et al., 2017). Several prior works examine the adversarial robustness of sparse networks (Sehwag et al., 2020; Fu et al., 2021). Simultaneously, work such as Stutz et al. (2021) suggests that adversarially robust networks have flatter minima.

In this work we focus on robustness to corrupted images (Hendrycks & Dietterich, 2019; Recht et al., 2019) due to its real-world applications such as autonomous driving. Diffenderfer et al. (2021) shows that sparse subnetworks can outperform the original dense model on clean and robust accuracy. However, almost all current pruning methods still lead to worse performance on OOD data than dense networks, a result we replicate in Figure 1 (Liebenwein et al., 2021; Hooker et al., 2019).

**Efficiency.** Efforts to improve network efficiency include pruning, distillation, quantization, and adaptation (Han et al., 2015a;b; Yu et al., 2019; Molchanov et al., 2016; 2019; 2022; Shen et al., 2022a; Hinton et al., 2015; Gholami et al., 2021; Banner et al., 2018; Dai et al., 2019; Yang et al., 2018). Two broad categories of pruning are structured and unstructured pruning. While unstructured pruning removes individual weights and can retain strong performance at high sparsities, structured pruning removes larger elements in the network, such as convolutional channels, and often allows for direct size or latency reduction of the pruned model (Anwar et al., 2017; Molchanov et al., 2019; Shen et al., 2022a). Unstructured pruning dates back to Optimal Brain Damage, but the Lottery Ticket Hypothesis has incited recent interest in the topic and spurred developments such as SNIP and GraSP (Lee et al., 2019; Frankle & Carbin, 2018; Hoffmann et al., 2021; LeCun et al., 1989; Tartaglione et al., 2022; Wang et al., 2020a; Yu et al., 2022). In the current work, we focus on structured channel-wise pruning.

**Flat minima.** The relationship between flat minima and generalization has been studied for many years (Hochreiter & Schmidhuber, 1997; Keskar et al., 2016; Wu et al., 2017; Guo et al., 2020; Jiang et al., 2019; Wu et al., 2020; Bartoldson et al., 2020; Cha et al., 2021; Izmailov et al., 2018; Kaddour et al.). Flat minima are often correlated with improved generalization, but the investigations into the extent of this correlation and the underlying causal relationship are still ongoing. Also, flatness is metric-dependent and sharp minima can generalize well (Dinh et al., 2017). Sharpness-Aware Minimization (SAM) (Foret et al., 2020) is a way to optimize for finding flatter minima and leads to better generalization. Work which builds upon SAM improves its efficiency (Liu et al., 2022;

Du et al., 2022; Jiang et al., 2023), modifies the optimization for improved performance (Kwon et al., 2021), applies it for compression in NLP applications (Na et al., 2022) and as a pre-pruning optimizer (Zhou et al., 2023), and modifies the objective for unstructured pruning (Peste et al., 2022). Stochastic weight averaging (SWA) is another method that has been used to optimize for flatter minima (Izmailov et al., 2018; Cha et al., 2021; Kaddour et al.). The present work is inspired by SAM and its use in model compression (Foret et al., 2020; Na et al., 2022) but differs in its use of adaptive weight perturbations and its emphasis on robustness. In this work we use the formulation introduced in ASAM (Kwon et al., 2021) due to its improved performance over that of SAM.

## 3 THE AdaSAP METHOD

Flat minima are minima within relatively large regions of the parameter space that have low loss. It has previously been observed that optimizing for flatter minima improves generalization and adversarial robustness in deep networks and generalization in sparse networks (Foret et al., 2020; Stutz et al., 2021; Na et al., 2022). Based on these observations, we hypothesize that optimizing for flatter minima during pruning can also enhance robustness. The main objective of our design is to find flat minima in order to produce models that are simultaneously prunable and robust. We introduce a new method, Adaptive Sharpness-Aware Pruning (AdaSAP). Figure 2 and Algorithm 2 detail the following three step procedure:

---

**Algorithm 2** AdaSAP Pruning Procedure

**Require:** Pretrained model weights $\mathbf{w}$ partitioned into neurons $w_i$, pruning importance criteria $\phi$, $\rho$ bounds $(\rho_{\min}, \rho_{\max})$

Let `train_iter` = Algorithm 1

**for** epoch **in** warmup_epochs **do**          ▷ Adaptive Weight Perturbation
    **for** each iteration **do**
        Sample training batch $b$
        $\mathbf{w}$ = `train_iter` $(\mathbf{w}, b, \rho_{\min}, \rho_{\max})$
    **end for**
**end for**
**for** epoch **in** pruning_epochs **do**                    ▷ Pruning
    **for** each iteration **do**
        Sample training batch $b$
        $\mathbf{w}$ = `train_iter` $(\mathbf{w}, b, \rho_{\min}, \rho_{\max})$
        **if** iteration % prune_frequency = 0 **then**
            $\mathbf{s} = \phi(\mathbf{w})$                    ▷ Score all neurons
            `idxs = rank`$(s)[:$ `prune_num`$]$ ▷ Neurons with lowest score
            $\mathbf{w}_{\texttt{idxs}} = 0$                    ▷ Prune neurons
        **end if**
    **end for**
**end for**
**for** epoch **in** finetune_epochs **do**          ▷ Robustness encouragement
    **for** each iteration **do**
        Sample training batch $b$
        $\mathbf{w}$ = `train_iter` $(\mathbf{w}, b, \rho_{\min} = \rho_{\max} = $ `constant`$)$   ▷ SAM
    **end for**
**end for**
**return** $\mathbf{w}$
**Ensure:** Pruned and finetuned model

---

**(1) Adaptive Weight Perturbations**. During warmup, we ensure that the neurons that will be pruned, are regularized to lie in flat regions so that they will not cause the model's performance to suffer too much after pruning.

**(2) Neuron Removal**. We conduct structured channel pruning and remove neurons according to any pruning criteria. Here we use magnitude pruning (Han et al., 2015b). During pruning, the model will preserve much of its performance since unimportant neurons have already been situated in flatter loss regions.

**(3) Robustness Encouragement**. While training the pruned model to convergence, we enforce flat minima, this time uniformly (*i.e.*, non-adaptively) across the entire network to promote robustness. This step is compatible with any flatness-based optimizer. Here we use ASAM (Kwon et al., 2021).

### 3.1 ADAPTIVE WEIGHT PERTURBATION

**Flatness-informed pruning.** Our procedure performs gradient updates which are informed by the local flatness in order to best prepare the network for pruning. We adapt the regularization per

neuron based on the importance, or likelihood of that neuron to be pruned. Pruning can be viewed as a special case of directional robustness, in which the weight and activation of a pruned neuron are set to zero. When the loss landscape is flat, the pruning process will incur only a small impact on the performance of the model. This formulation indicates that pruning and robustness share a common characterization through the perspective of sharpness, as demonstrated in Figure 3.

Taking inspiration from the uniform $\epsilon$-ball perturbation based method in Foret et al. (2020) and its scale-invariant version in Kwon et al. (2021), among several other approaches for achieving flatness in dense models (Izmailov et al., 2018; Santurkar et al., 2018), we perturb neurons according to their importance to better prepare the network for pruning. We follow the intuition that important weights are worth adding some sharpness to our model, while unimportant weights can have stronger regularization. This idea is supported by Molchanov et al. (2019), which shows that pruning neurons in flat regions hurts the loss less than pruning neurons in sharp regions. We adapt the size of allowable perturbations based on the likelihood of each weight to be pruned: weights with low importance scores have large perturbation balls which enforce that they lie within flatter regions and vice versa.

**Setup.** Following this intuition, we derive the gradient update for training. Consider a network whose parameters are grouped into $I$ partitions, where each partition $i$ includes the set of weights $\mathbf{w}_i$. In our setting, since we consider channel-wise pruning, each channel constitutes a partition. For simplicity, throughout this paper we refer to a channel-wise group of weights as a *neuron*. We use $\mathbf{w}_i$ to refer to a single neuron and $\mathbf{w}$ to refer to the collection of all neurons $\{\mathbf{w}_i\}_{i=1}^I$.

**Perturbation regions.** Consider training loss $L_S(\mathbf{w})$ of the network over the training dataset $S$. Consider a set of weight perturbation ball radius sizes $\boldsymbol{\rho}$, with each $\rho_i$ corresponding to the size of allowable perturbations for neuron $\mathbf{w}_i$. Our goal is to optimize for a network in which unimportant neurons lie in flatter minima prior to pruning via the following objective

$$\min_{\mathbf{w}} \max_{\{\epsilon : \|T_{\mathbf{w}}^{-1}\epsilon_i\| \leq \rho_i\}_{i=1}^I} L_S(\mathbf{w} + \epsilon). \tag{1}$$

where $T_{\mathbf{w}}$ and its inverse $T_{\mathbf{w}}^{-1}$ are transformations that can be applied during optimization so as to reshape the perturbation region (*i.e.*, not necessarily a ball). Kwon et al. (2021) showed that this strategy allows for greater perturbation exploration and leads to improved performance. $\rho_i$ values are based on computing an importance score for each neuron, similar to a pruning criterion. Consider a neuron importance score function $\psi(\cdot)$, such as the $\ell_2$ norm of the neuron weight or other scores that are some function of weights, gradients, and higher order terms (Molchanov et al., 2019; Lee et al., 2019; Yu et al., 2022; Shen et al., 2022a; Wang et al., 2020a). Given each neuron score $s_i = \psi(\mathbf{w}_i)$, we can now compute the perturbation ball size $\rho_i$, which projects $s_i$ to lie within $(\rho_{min}, \rho_{max})$

$$\rho_i = \rho_{\max} - \frac{s_i - s_{\min}}{s_{\max} - s_{\min}}(\rho_{\max} - \rho_{\min}). \tag{2}$$

Scores $s_{\min}$ and $s_{\max}$ are obtained empirically at each gradient step, but $\rho_{\max}$ and $\rho_{\min}$ are hyperparameters that are set ahead of time and we find them to scale across various experiment settings (further details in Section 4 and Appendix A). Neuron importance estimates $\psi(\cdot)$ can be evaluated during training with little additional computational overhead.

**Optimal gradient update.** Our enhanced gradient update optimizes for finding a minimum that also lies in a region of low loss. Beginning with our objective in Eq. 1, we perform a series of approximations in order to find the most adversarial perturbations $\boldsymbol{\epsilon} = \{\epsilon_i\}$ (*i.e.*, highest loss near the minimum) which we can apply to the neurons $\mathbf{w} = \{\mathbf{w}_i\}$ in our network in order to obtain a flatness-informed gradient update.

We use a first order Taylor expansion around each neuron $i$ with respect to $\epsilon$ around $0$ (in line with Foret et al. (2020)) to approximate the inner maximization in Eq. 1. This gives us $\epsilon^*$, the optimal $\epsilon$ we can use to inform our gradient update. Define $\tilde{\epsilon}_i = T_{\mathbf{w}}^{-1}\epsilon_i$. For simplicity we denote the conditional loss due to weight perturbations as the function $L_{S,\mathbf{w}}(\cdot)$. Then we derive our optimal update as follows for each neuron:

$$\begin{aligned} \tilde{\epsilon}_i^* &= \arg\max_{\|\tilde{\epsilon}_i\|_2 \leq \rho_i} L_{S,\mathbf{w}}(\mathbf{w}_i + T_{\mathbf{w}}\tilde{\epsilon}_i), \\ &\approx \arg\max_{\|\tilde{\epsilon}_i\|_2 \leq \rho_i} L_{S,\mathbf{w}}(\mathbf{w}_i) + \tilde{\epsilon}_i^\top T_{\mathbf{w}} \nabla_{\mathbf{w}_i} L_{S,\mathbf{w}}(\mathbf{w}_i), \\ &= \arg\max_{\|\tilde{\epsilon}_i\|_2 \leq \rho_i} \tilde{\epsilon}_i^\top T_{\mathbf{w}} \nabla_{\mathbf{w}_i} L_{S,\mathbf{w}}(\mathbf{w}_i), \end{aligned} \tag{3}$$

$\| \cdot \|_2$ being the $\ell_2$ norm. We can then approximate $\tilde{\epsilon}^*$ by rescaling the gradient associated with each neuron so that its norm is $\rho_i$ through

$$\tilde{\epsilon}_i^* = \rho_i \frac{\text{sign}(\nabla_{w_i} L_{S,\mathbf{w}}(\mathbf{w}_i))|T_{\mathbf{w}} \nabla_{\mathbf{w}_i} L_{S,\mathbf{w}}(\mathbf{w}_i)|}{(\|T_{\mathbf{w}} \nabla_{\mathbf{w}_i} L_{S,\mathbf{w}}(\mathbf{w}_i)\|_2^2)^{1/2}}. \tag{4}$$

Since we were originally approximating $\tilde{\epsilon}_i = T_{\mathbf{w}}^{-1} \epsilon_i$, we recover our approximation of $\epsilon^*$ as

$$\hat{\epsilon}_i = \rho_i \frac{T_{\mathbf{w}}^2 \nabla_{\mathbf{w}_i} L_{S,\mathbf{w}}(\mathbf{w}_i)}{\|\nabla_{\mathbf{w}_i} L_{S,\mathbf{w}}(\mathbf{w}_i)\|_2}. \tag{5}$$

**Approximating the optimal gradient update.** We then find an approximation to our desired gradient. Referring to Eq. 1, we can find the gradient of the inner maximization in order to derive a sharpness-aware gradient update. We use the inner maximization and plug in our approximation $\hat{\epsilon}$ for each individual neuron during adjacent pruning steps, defined as

$$\nabla_{\mathbf{w}_i} \max_{\|T_{\mathbf{w}}^{-1} \epsilon_i\|_2 \leq \rho_i} L_{S,\mathbf{w}}(\mathbf{w}_i + \epsilon_i) \approx \nabla_{\mathbf{w}_i} L_{S,\mathbf{w}}(\mathbf{w}_i + \hat{\epsilon}_i)$$

$$= \frac{d(\mathbf{w}_i + \hat{\epsilon}_i)}{d\mathbf{w}_i} \nabla_{\mathbf{w}_i} L_{S,\mathbf{w}}(\mathbf{w}_i)|_{\mathbf{w}_i + \hat{\epsilon}_i}$$

$$= \nabla_{\mathbf{w}_i} L_{S,\mathbf{w}}(\mathbf{w}_i)|_{\mathbf{w}_i + \hat{\epsilon}_i} + \frac{d\hat{\epsilon}_i}{d\mathbf{w}_i} \nabla_{\mathbf{w}_i} L_{S,\mathbf{w}}(\mathbf{w}_i)|_{\mathbf{w}_i + \hat{\epsilon}_i}$$

$$\approx \nabla_{\mathbf{w}_i} L_{S,\mathbf{w}}(\mathbf{w}_i)|_{\mathbf{w}_i + \hat{\epsilon}_i}, \tag{6}$$

where we drop the second order terms for efficiency. This leads us to our final gradient update approximation between adjacent pruning steps as $\nabla_{\mathbf{w}_i} L_{S,\mathbf{w}}(\mathbf{w}_i) \approx \nabla_{\mathbf{w}_i} L_{S,\mathbf{w}}(\mathbf{w}_i)|_{\mathbf{w}_i + \hat{\epsilon}_i}$.

We use this gradient update in place of the standard gradient update during the warmup phase of our pruning procedure before neuron removal. This sets up our network so that when pruning begins, we have a model that is closer to a flat minima. Optimization details are in Algorithm 1.

## 3.2 NEURON REMOVAL

We focus on structured (channel-wise) pruning since it leads to models that can take advantage of direct computational resource savings on GPUs, leading to inference speedup (Molchanov et al., 2022; Shen et al., 2022a; Yang et al., 2021).

In this stage of the procedure, we remove unimportant neurons according to any scoring function $\phi(\cdot)$ that measures neuron saliency. This may be the same or different from the scoring function $\psi(\cdot)$ used in determining adaptive weight perturbation ball sizes during the first step. AdaSAP works with a range of pruning methods as we show later in Section 4.

## 3.3 ROBUSTNESS ENCOURAGEMENT

In the third and final phase, we focus on optimizing our model for robust performance. We regularize our weights uniformly since we have completed pruning and now want to enforce robustness for the entire network. This is illustrated in Figure 3 in the plot on the right: the network loss relative to a flatter neuron will stay more stable in the presence of corruptions, whereas the loss relative to a sharper neuron could fluctuate significantly. This portion of the procedure could be instantiated with any sharpness-based method that optimizes for an overall flatter minima.

## 3.4 FINAL COMMENTS

We emphasize that AdaSAP *is not a pruning method* but rather an optimization paradigm that performs robustness-aware pruning. Our method can be used in conjunction with any existing pruning criteria. In Section 4 we show that AdaSAP (in conjunction with magnitude pruning) generally outperforms existing SOTA pruning methods. As new pruning techniques arise, they could be enhanced via AdaSAP in order to obtain increased performance, particularly robust performance.

Additionally, we note that our method *differs from robust pruning methods* such as Sehwag et al. (2020); Vemparala et al. (2021); Zhao & Wressnegger (2022) since we consider a setting in which

the model may be exposed to *novel types of corruptions at test time*. Therefore, we do not want to include, i.e. ImageNet C or ImageNet V2 images in our training set. We demonstrate results on these OOD datasets in an attempt to demonstrate the versatility of our method: despite only training on ImageNet data, we can still expect the models to be robust to unseen input variation.

# 4  EXPERIMENTS & RESULTS

## 4.1  EXPERIMENT DETAILS

We describe the basics of our experimental setup here. Additional experiment details and confidence intervals can be found in Appendix A.

**Datasets.** For image classification, we train on ImageNet-1K (Deng et al., 2009) and additionally evaluate on ImageNet-C (Hendrycks & Dietterich, 2019) and ImageNet-V2 (Recht et al., 2019). For object detection, we use the Pascal VOC dataset (Everingham et al., 2009). To assess robustness, we create a Pascal VOC-C dataset by applying ImageNet-C-style corruptions to the test set.

**Pruning criteria.** Our proposal is agnostic to the particular importance criteria $\phi$ used to prune the network. We evaluate our method on two types of pruning: *parameter-based* and *latency-based*. We choose to group our methods into these two categories due to the different goals of pruning methods. Some methods, such as magnitude pruning, seek to reduce the number of parameters to produce a small model. Other methods, such as HALP (Shen et al., 2022b), seek to produce a model with the fastest latency speedup, regardless of the number of parameters.

In both cases, we perform structured pruning, that is, pruning channels rather than individual parameters. We refer to our two methods as $AdaSAP_P$ and $AdaSAP_L$, to denote the parameter-specific method and the latency-specific method, respectively. We use $\ell_2$ norm magnitude pruning for $AdaSAP_P$ and HALP for $AdaSAP_L$. Details of the pruning schedule are in Appendix A.

**Metrics.** For image classification, we report the Top1 accuracy on each dataset and two *robustness ratios*, defined as the ratio in robust accuracy to validation accuracy: $R_C = \mathrm{acc}_C/\mathrm{acc}_{val}$ and $R_{V2} = \mathrm{acc}_{V2}/\mathrm{acc}_{val}$. Improvement in these ratios indicates that we are closing the gap between robust and validation performance. For object detection, we evaluate the model on Pascal VOC and VOC-C. We report mean average precision (mAP) and a robustness ratio $R_C = \mathrm{mAP}_C/\mathrm{mAP}_{val}$.

**Baselines.** We primarily focus on comparing to two SOTA baselines which we re-run in our experiments, Taylor pruning for parameter-based pruning (Molchanov et al., 2019), and HALP for latency-based pruning (Shen et al., 2022a). Taylor pruning assigns a score to each neuron based on a Taylor series approximation of how much the loss would change if the neuron were removed. HALP is a pruning method that optimizes for improving latency. We additionally run results on magnitude pruning (Han et al., 2015b), to demonstrate the improvements of $AdaSAP_P$ over its closest baseline. We also compare to the results cited in several other pruning methods such as GR-eg (Wang et al., 2020b), EagleEye (Li et al., 2020), ABCPruner (Lin et al., 2020), and SMCP (Humble et al., 2022).

## 4.2  IMAGENET CLASSIFICATION

**Parameter reduction.** Table 1 shows how AdaSAP compares favorably to other pruning methods at a variety of pruning ratios for ResNet50. MobileNet V1 and V2 results show a similar trend and are included in Appendix B. First, AdaSAP has the best overall $R_C$ and $R_{V2}$ ratios among all pruning methods. This means that our method helps to close the gap between robust performance and standard validation performance. Additionally, our method consistently outperforms comparisons on both ImageNet validation performance and on robust performance (via ImageNet C and ImageNet V2) while at the same or even smaller compressed model size.

**Latency Reduction.** In a second set of experiments, we focus on latency constraints. To this end, we consider the most recent hardware-aware pruning method HALP (NeurIPS'22 (Shen et al., 2022b)), which incorporates latency into the metric to measure the saliency of each neuron. Table 2 shows that using $AdaSAP_L$ outperforms prior works both in terms of the $R_C$ and $R_{V2}$ robustness ratios, and also in terms of standard validation performance and performance on OOD datasets.

Table 1: **ResNet50 - Parameter Reduction.**

| Method | Size ↓ | Val | $R_{V2}$ | $R_C$ | IN-V2 | IN-C |
|---|---|---|---|---|---|---|
| | | **ResNet50** | | | | |
| Dense | 1 | 77.32 | 0.83 | 0.55 | 64.79 | 42.46 |
| Magnitude | 0.20 | 73.80 | 0.83 | 0.48 | 61.40 | 35.27 |
| Taylor | 0.20 | 73.56 | 0.82 | 0.46 | 60.56 | 33.93 |
| EagleEye 1G | 0.27 | 74.13 | 0.83 | 0.47 | 61.30 | 34.84 |
| **AdaSAP**$_P$ | **0.20** | **74.63** | **0.83** | **0.50** | **62.08** | **37.30** |
| Magnitude | 0.44 | 76.83 | **0.85** | 0.51 | **64.92** | 39.27 |
| Taylor | 0.42 | 75.85 | 0.84 | 0.50 | 63.51 | 37.84 |
| Greg-1 | 0.43 | 73.72 | 0.82 | 0.48 | 60.47 | 35.04 |
| Greg-2 | 0.43 | 73.84 | 0.83 | 0.48 | 61.05 | 35.39 |
| ABCPruner | 0.44 | 73.52 | 0.82 | 0.50 | 60.46 | 36.64 |
| **AdaSAP**$_P$ | **0.40** | **77.27** | 0.83 | **0.53** | 64.51 | **41.23** |
| Greg-1 | 0.62 | 75.16 | 0.82 | 0.49 | 61.82 | 36.88 |
| Greg-2 | 0.62 | 75.36 | 0.82 | 0.49 | 62.06 | 37.09 |
| ABCPruner | 0.71 | 74.84 | 0.83 | 0.51 | 61.73 | 38.35 |
| **AdaSAP**$_P$ | **0.62** | **77.99** | **0.84** | **0.52** | **65.49** | **42.68** |
| Magnitude | 0.76 | 77.32 | 0.84 | 0.53 | 65.20 | 40.73 |
| Taylor | 0.76 | 77.05 | 0.84 | 0.52 | 64.53 | 39.68 |
| Greg-1 | 0.77 | 76.25 | 0.83 | 0.51 | 63.61 | 38.96 |
| EagleEye 3G | 0.78 | 77.07 | 0.84 | 0.53 | 64.84 | 40.67 |
| **AdaSAP**$_P$ | **0.77** | **78.29** | **0.84** | **0.55** | **66.14** | **43.22** |

Table 2: **ResNet50 - Latency Reduction.**

| Method | Speedup ↑ | Size ↓ | Val | $R_{V2}$ | $R_C$ | IN-V2 | IN-C |
|---|---|---|---|---|---|---|---|
| | | | **ResNet50** | | | | |
| Dense | 1 | 1 | 77.32 | 0.838 | 0.55 | 64.79 | 42.46 |
| HALP | 2.6 | 0.43 | 74.46 | 0.82 | 0.47 | 61.21 | 35.03 |
| **AdaSAP**$_L$ | 2.6 | 0.41 | **75.37** | **0.83** | **0.50** | **62.61** | **37.93** |
| HALP | 1.6 | 0.68 | 76.55 | 0.83 | 0.51 | 63.62 | 39.13 |
| SMCP | 1.7 | 0.60 | 76.62 | 0.83 | 0.51 | 63.86 | 38.72 |
| **AdaSAP**$_L$ | 1.6 | 0.65 | **77.28** | **0.85** | **0.54** | **65.35** | **41.63** |
| HALP | 1.2 | 0.87 | 77.45 | 0.84 | 0.53 | 64.88 | 40.77 |
| SMCP | 1.2 | 0.87 | 77.57 | 0.84 | 0.53 | 65.02 | 40.91 |
| **AdaSAP**$_L$ | 1.1 | 0.82 | **77.93** | 0.84 | **0.55** | **65.53** | **42.92** |

## 4.3 OBJECT DETECTION

We show object detection results in Table 3. Across two pruning levels, AdaSAP consistently out-performs the HALP baseline by noticeable margins. More details can be found in Appendix A.

## 4.4 ABLATIONS

**Sensitivity to importance metrics.** In our experiments, we considered $\ell_2$ norm, or magnitude, as the saliency metric for pruning since this led to the best performance. However, our method is flexible enough to work with other metrics such as Taylor importance as seen in Table 5.

**Adaptive perturbations.** In order to determine the importance of using adaptive weight perturbations, we compare our method to the use of uniform perturbations throughout all three steps, which is equivalent to using SAM. In Table 6, we compare three different setups: Taylor pruning with SGD as a baseline, Taylor pruning with SAM, and AdaSAP$_P$. All three settings are fine-tuned for 90 epochs. Our method outperforms both SGD and SAM at the same or smaller compressed size. This demonstrates that our method's adaptive perturbations during the warmup phase are helping the network achieve a more prunable state. See Appendix C for more extensive results and further analysis of the choice of uniform perturbations during robustness encouragement.

Table 3: **Object Detection.** mAP on validation images and ImageNetC style corruptions on the Pascal VOC dataset.

| Method | Size ↓ | Val | Corrupted | $R_C$ |
|---|---|---|---|---|
| HALP | 0.40 | 0.774 | 0.583 | 0.753 |
| **AdaSAP** | 0.40 | **0.795** | **0.620** | **0.780** |
| HALP | 0.20 | 0.770 | 0.580 | 0.753 |
| **AdaSAP** | 0.20 | **0.793** | **0.616** | **0.776** |

Table 5: **AdaSAP can work with any pruning criteria.** AdaSAP with Taylor pruning matches or outperforms SGD with Taylor pruning.

| Method | Size ↓ | Val | $R_C$ | IN-C |
|---|---|---|---|---|
| Taylor + SGD | 0.42 | 75.85 | 0.50 | 37.84 |
| AdaSAP$_{P,\text{Taylor}}$ | 0.43 | 76.26 | 0.50 | 38.07 |
| **AdaSAP$_P$** | **0.41** | **76.93** | **0.52** | **39.64** |
| Taylor + SGD | 0.76 | 77.05 | 0.52 | 39.68 |
| AdaSAP$_{P,\text{Taylor}}$ | 0.76 | 77.42 | 0.52 | 40.27 |
| **AdaSAP$_P$** | **0.76** | **77.86** | **0.53** | **41.30** |

Table 4: **Sharpness before and after pruning without finetuning**. Lower sharpness values indicates a flatter loss landscape. AdaSAP achieves flatter minima both directly before and after pruning, before any finetuning.

| Method | Size ↓ | Sharpness pre pruning | Sharpness post pruning |
|---|---|---|---|
| Taylor | 0.42 | 0.039 | 0.044 |
| **AdaSAP$_P$** | **0.40** | **0.037** | **0.039** |
| Taylor | 0.76 | 0.039 | 0.041 |
| **AdaSAP$_P$** | **0.76** | **0.037** | **0.038** |

Table 6: **Adaptive perturbations are critical for the best clean and robust performance.**

| Method | Size ↓ | Val | $R_C$ | IN-C |
|---|---|---|---|---|
| **SGD** | | | | |
| Dense | 1 | 77.32 | 0.54 | 42.46 |
| Taylor + SGD | 0.20 | 73.56 | 0.46 | 33.93 |
| **AdaSAP vs. SAM (without ASAM)** | | | | |
| Taylor + SAM | 0.20 | 73.62 | 0.47 | 34.49 |
| **AdaSAP$_P$** | 0.20 | **74.38** | **0.48** | **35.86** |
| **AdaSAP vs. SAM (with ASAM)** | | | | |
| SAM + ASAM | 0.19 | 73.93 | 0.50 | 36.66 |
| **AdaSAP$_P$ + ASAM** | 0.19 | **74.63** | 0.50 | **37.30** |

### 4.5 SHARPNESS ANALYSIS

Consider the sharpness defined in Foret et al. (2020) as $\max_{\|\epsilon\|_2 \leq \rho} L_S(\mathbf{w} + \epsilon) - L_S(\mathbf{w})$. This measures the maximum amount that the loss could change if the weights are perturbed within a ball of radius $\rho$. We measure sharpness directly before and after pruning, to evaluate how sharpness prepares the network to be optimally pruned, as well as how pruning affects the model sharpness. Table 4 shows that using AdaSAP leads to flatter models both before and after pruning. In line with previous results that found an association between flatness and generalization, this result can help explain why our models have better generalization and robust generalization performance.

### 4.6 LIMITATIONS

One limitation is that AdaSAP requires twice the training time due to the two backward passes necessary for optimization. A 90 epoch pruning and finetuning run on 8 V100 GPUs to reduce a ResNet50 network to $0.2\times$ its size takes takes 32 hours and 12 minutes for AdaSAP$_P$ while SGD with magnitude pruning takes 16 hours and 53 minutes. Recent strategies (Du et al., 2022; Liu et al., 2022; Jiang et al., 2023) could be used to reduce this additional overhead. Another limitation is that our analysis is based on existing robustness datasets that don't fully capture the real world corruptions encountered in autonomous driving. Despite a focus in the present work on applying pruning (and fine-tuning) to an existing high performing dense model, AdaSAP shows promise in also being used to prune models from scratch. Finally, although we focus here on the more challenging structured pruning, we believe AdaSAP could also benefit unstructured pruning.

## 5 CONCLUSION

We introduce the Adaptive Sharpness-Aware Pruning method (AdaSAP) which optimizes for both accuracy and robust generalization during the pruning procedure. The method consists of three steps: application of our novel adaptive weight perturbations, pruning, and flatness-based robustness encouragement. AdaSAP outperforms a variety of SOTA pruning techniques on clean and robust performance and relative robustness in both image classification and object detection.

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

# A EXPERIMENT DETAILS

## A.1 DATASETS

We conduct image classification experiments on the ImageNet dataset (Deng et al., 2009). To evaluate the generalization ability of pruned models, we also evaluate models on ImageNet-C (Hendrycks & Dietterich, 2019) and ImageNet-V2 (Recht et al., 2019) datasets. The former consists of the validation images from the ImageNet dataset, but with nineteen types of corruptions applied with five different levels of severity. The latter is a dataset created in the same manner as the original dataset, but consisting of different images. Therefore, it is intended to measure out of distribution performance of models. This ImageNet-V2 dataset includes three different sets, each one with a slightly different sampling strategy. We focus our experiments on the MatchedFrequencies dataset, but include evaluation on all three datasets in Table 8 and see that our method consistently outperforms the baseline. These results do not use the ASAM formulation.

We conduct object detection experiments on the Pascal VOC dataset (Everingham et al., 2009). We additionally create a Pascal VOC-C dataset, in which we apply the ImageNet-C corruptions to the Pascal VOC test set. We only use one severity level.

## A.2 ADDITIONAL ROBUSTNESS DATASETS

As described above, we choose the ImageNet V2 Matched Frequencies dataset to perform our main set of experiments, but there are two other ImageNet V2 datasets: Threshold 0.7 and Top Images. These three datasets vary in terms of selection criteria for included images. Matched Frequencies attempts to match the image selection frequency of MTurk workers in the original ImageNet validation set. Threshold 0.7 samples from among images with a selection frequency of at least 0.7. Top Images includes the images with the highest selection frequency within each class. We provide the pruned model robustness comparison on these the additional ImageNet V2 datasets in Table 8.

## A.3 NETWORK ARCHITECTURES

In our experiments we prune three different networks for the classification task: ResNet50, MobileNet V1 and MobileNet V2 (He et al., 2016; Howard et al., 2017). All ablation studies use ResNet50 with ImageNet and ImageNet C datasets unless otherwise specified. "Val" column labels refer to standard ImageNet validation accuracy or loss. We use a pretrained model trained for 90 epochs with a cosine learning rate as in HALP (Shen et al., 2022b) and EagleEye (Li et al., 2020). For object detection, we follow the experimental setup in HALP (Shen et al., 2022b) to prune an SSD512 model with ResNet50 as the backbone. We perform Distributed Data Parallel training across 8 V100 GPUs with batch size 128 for all experiments.

## A.4 PRUNING SCHEDULE

Given a pre-trained model, for any architecture, we run the warm up for 10 epochs, and then we follow the same pruning schedule as in de Jorge et al. (2020): we prune every 30 iterations and, in each iteration, we prune away a $p_r$ fraction of neurons so that the final network is pruned by a fraction $p$ (resulting in a network of size $1 - p$). To determine the $p_r$ fraction, we follow an exponential decay schedule. Let $k = 1 - p$ and let $k_r$ be the number of neurons remaining after $r$ pruning iterations, where the total number of pruning iterations is $R$. Let $m$ be the number of neurons in the dense network, and define $\alpha = \frac{r}{R}$. Then, $k_r = \exp\{\alpha \log k + (1 - \alpha \log m)\}$. We fine-tune the pruned model for another 79 epochs (to reach 90 epochs total).

## A.5 OPTIMIZATION HYPERPARAMETERS

The base optimizer is SGD with cosine annealing learning rate with a linear warmup over 8 epochs, a largest learning rate of $1.024$, momentum of $0.875$, and weight decay $3.05e - 05$. Unless otherwise stated we use $\rho_{min} = 0.01$ and $\rho_{max} = 2.0$ for all experiments – we observe these values scale well across networks and tasks. For robustness encouragement we use $\rho = 2.0$ in line with prior work (Kwon et al., 2021). For some ablation experiments, the original SAM (Foret et al., 2020)

optimizer is sufficient and in these cases we reduce $\rho_{max} = 0.1$ and use constant $\rho = 0.05$ for finetuning. We include some simple hyperparameter tuning results for $\rho$ values in Table 7.

Table 7: Simple hyperparameter tuning results on ResNet50 model.

| $\rho_{min}$ | $\rho_{max}$ | Epoch 2 Val Loss | Epoch 2 Val Acc |
|---|---|---|---|
| **0.01** | **0.05** | **5.54** | **8.63** |
| 0.01 | 0.1 | 5.59 | 7.11 |
| 0.05 | 0.1 | 5.73 | 5.69 |
| 0.05 | 0.5 | 6.92 | 0.1 |

Table 8: Our AdaSAP method outperforms the Taylor pruning baseline on the additional ImageNet V2 datasets. Additionally, the AdaSAP model pruned to 76% outperforms the dense model across all three datasets. Results are on ResNet50 and do not use ASAM.

| Method | Size ↓ | Matched Frequences | Threshold 0.7 | Top Images |
|---|---|---|---|---|
| Dense | 1 | 64.80 | 73.79 | 79.00 |
| Taylor | 0.20 | 60.56 | 69.74 | 75.53 |
| **AdaSAP**$_P$ | 0.20 | **62.03** | **70.83** | **76.62** |
| Taylor | 0.40 | 63.51 | 72.54 | 77.83 |
| **AdaSAP**$_P$ | 0.40 | **64.62** | **73.75** | **78.87** |
| Taylor | 0.76 | 64.53 | 73.32 | 78.72 |
| **AdaSAP**$_P$ | 0.76 | **66.00** | **74.70** | **79.66** |

## B  ADDITIONAL RESULTS

### B.1  MOBILENET RESULTS

We include results on MobileNet V1 and MobileNet V2 in Tables 9 and 10. These parallel our main results in the main paper, in which we evaluated AdaSAP$_P$ and AdaSAP$_L$ on ResNet50 and compared them against other pruning methods. Here, we perform a similar comparison, where Table 9 includes our results on AdaSAP$_P$ and Table 10 includes our results on AdaSAP$_L$. We compare against Taylor importance (Molchanov et al., 2019), EagleEye (Li et al., 2020), and PolarReg (Zhuang et al., 2020) for AdaSAP$_P$ and against HALP (Shen et al., 2022b), SMCP (Humble et al., 2022), MetaPruning (Liu et al., 2019), AutoSlim (Yu & Huang, 2019), AMC (He et al., 2018), and EagleEye (Li et al., 2020) for AdaSAP$_L$. Results here do not use ASAM. We can observe that AdaSAP performs strongly compared to baselines, particularly in the parameter-based setting.

### B.2  MARGIN OF IMPROVEMENT IN CLASSIFICATION

Figure 4 shows the margin of performance on various corruption types for the classification task. AdaSAP outperforms a Taylor pruned model on all corruptions, across three different sparsities. We can see that it tends to particularly improve performance on several corruptions that may be important for the autonomous driving application, such as pixelated images, fog, and snow.

### B.3  MARGIN OF IMPROVEMENT IN OBJECT DETECTION

Similarly to our result on classification, we include margins of performance improvement on various corruption types for the object detection task in Figure 5.

### B.4  RELATIVE ROBUSTNESS

In Figure 6 we show that AdaSAP outperforms baselines on each of the constituent elements of the relative robustness metric. Recall that relative robustness is robust accuracy divided by standard

validation accuracy. In addition to outperforming baselines on relative robustness, AdaSAP also outperforms on ImageNet validation accuracy and ImageNet C robust accuracy.

Table 9: **MobileNet-V1/V2 - Parameters.** Top1 Accuracy as a function of the pruning ratio.

| Method | Size ↓ | Val | $R_{V2}$ | $R_C$ | IN-V2 | IN-C |
|---|---|---|---|---|---|---|
| **MobileNet-V1** | | | | | | |
| Dense | 1 | 72.63 | 0.82 | 0.45 | 59.30 | 32.79 |
| Taylor | 0.40 | 69.61 | 0.80 | 0.42 | 55.90 | 29.51 |
| **AdaSAP**$_P$ | **0.39** | **71.05** | **0.82** | **0.44** | **58.08** | **31.06** |
| Taylor | 0.52 | 71.21 | 0.81 | 0.43 | 57.65 | 30.92 |
| **AdaSAP**$_P$ | **0.52** | **71.58** | **0.82** | **0.44** | **58.75** | **31.77** |
| EagleEye | 0.56 | 70.86 | 0.80 | 0.42 | 56.88 | 29.98 |
| **AdaSAP**$_P$ | **0.56** | **72.05** | **0.82** | **0.45** | **58.94** | **32.24** |
| **MobileNet-V2** | | | | | | |
| Dense | 1 | 72.10 | 0.81 | 0.45 | 58.50 | 32.40 |
| Taylor | 0.72 | 70.49 | 0.81 | 0.43 | 56.87 | 30.22 |
| **AdaSAP**$_P$ | **0.72** | **72.06** | **0.82** | **0.45** | **58.73** | **32.63** |
| PolarReg | 0.87 | 71.72 | 0.82 | 0.45 | 58.46 | 32.04 |
| Taylor | 0.88 | 71.93 | 0.82 | 0.45 | 58.75 | 32.21 |
| **AdaSAP**$_P$ | 0.88 | **72.34** | **0.82** | **0.45** | **59.28** | **32.80** |

Table 10: **MobileNet-V1/V2 - Latency.** Top1 accuracy for latency constrained pruning for various speedup ratios. "–" indicates that we could not evaluate the model due to unavailable code or models.

| Method | Speedup ↑ | Val | $R_{V2}$ | $R_C$ | IN-V2 | IN-C |
|---|---|---|---|---|---|---|
| **MobileNet-V1** | | | | | | |
| Dense | 1 | 72.63 | 0.82 | 0.45 | 59.30 | 32.79 |
| MetaPruning | 2.06 | 66.1 | – | – | – | – |
| AutoSlim | 2.27 | 67.9 | – | – | – | – |
| HALP | 2.32 | 68.30 | 0.80 | 0.41 | 54.95 | 28.15 |
| SMCP | **2.39** | 68.34 | 0.80 | **0.42** | 54.38 | **28.68** |
| **AdaSAP**$_L$ | 2.33 | **68.45** | **0.81** | 0.41 | **55.42** | 28.29 |
| 0.75 MobileNetV1 | 1.37 | 68.4 | – | – | – | – |
| AMC | 1.42 | 70.5 | – | – | – | – |
| MetaPruning | 1.42 | 70.9 | – | – | – | – |
| EagleEye | 1.47 | 70.86 | 0.80 | 0.42 | 56.88 | 29.98 |
| HALP | 1.68 | 71.31 | 0.81 | 0.43 | 57.38 | 30.77 |
| SMCP | **1.72** | 71.00 | 0.81 | 0.44 | 57.20 | 31.02 |
| **AdaSAP**$_L$ | 1.70 | **71.48** | **0.82** | **0.44** | **58.23** | **31.35** |
| **MobileNet-V2** | | | | | | |
| Dense | 1 | 72.10 | 0.81 | 0.45 | 58.50 | 32.40 |
| HALP | **1.84** | 70.42 | 0.81 | 0.45 | 57.21 | 31.69 |
| **AdaSAP**$_L$ | 1.81 | **71.35** | 0.81 | **0.46** | **57.85** | **32.63** |
| HALP | 1.33 | 72.16 | 0.81 | 0.46 | 58.53 | **33.04** |
| **AdaSAP**$_L$ | **1.39** | **72.19** | **0.82** | 0.46 | **59.36** | 32.91 |

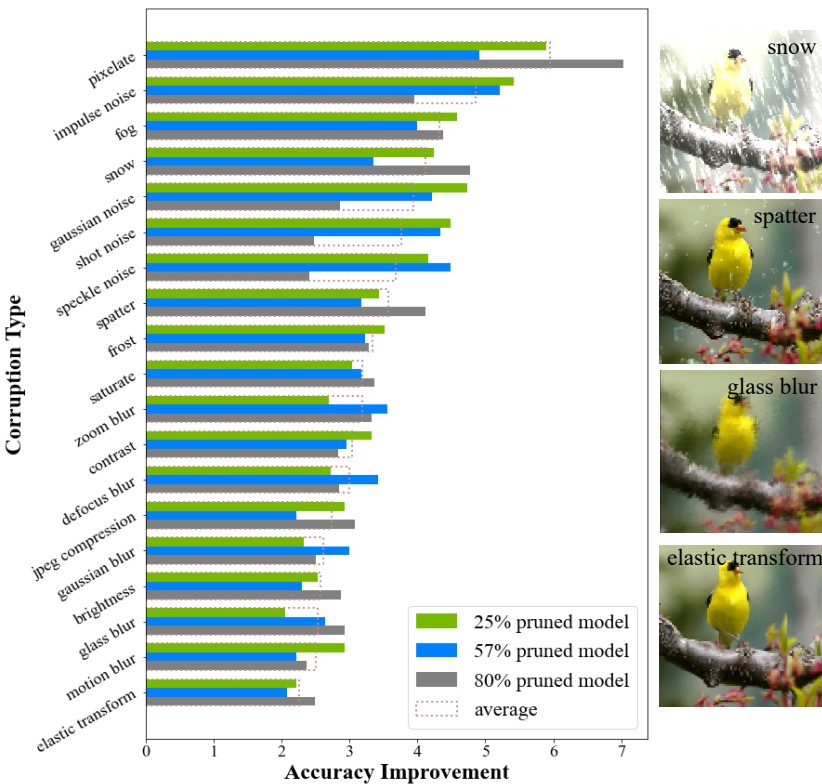

Figure 4: **ImageNet C Dataset.** Performance difference on various ImageNet C corruption types on models of varying sparsity. Accuracy improvement is the Top1 accuracy on a ResNet50 model trained and pruned using AdaSAP minus that of a Taylor pruned model.

## B.5 ADDITIONAL SHARPNESS METRICS

We additionally run limited measurements of an alternative sharpness metric: the top eigenvalue of the Hessian. We compare the sharpness of a model trained with AdaSAP vs. SGD in Table 11.

Table 11: Top Hessian eigenvalue sharpness measurements on ResNet50 model.

| Optimizer | Pre-pruning | Post-pruning | Post-finetuning |
|---|---|---|---|
| SGD | 35.91 | 93.82 | 8.17 |
| **AdaSAP**$_L$ | **25.23** | **16.79** | **7.66** |

## C ADDITIONAL ABLATIONS

### C.1 PERFORMANCE CHANGE AFTER PRUNING

In Table 14 we show how the loss and accuracy change over the course of pruning. Our hypothesis is that our method sets up the network for better pruning, so that the performance drop over the course of pruning is minimized. In most cases, our method has the best validation loss and accuracy both before and after pruning. This indicates that our method sets up the model to be pruned well, and also preserves performance well throughout the pruning process.

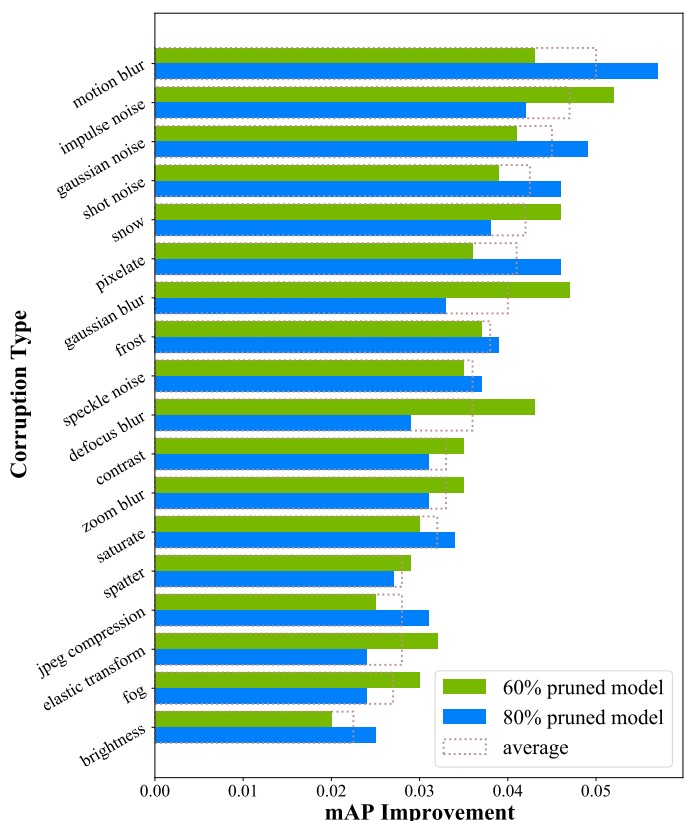

Figure 5: **Pascal VOC-C dataset.** Performance difference on various ImageNet C corruption types on models of varying sparsity. mAP improvement is the mAP of a ResNet50 model trained and pruned using AdaSAP minus that of a HALP pruned model.

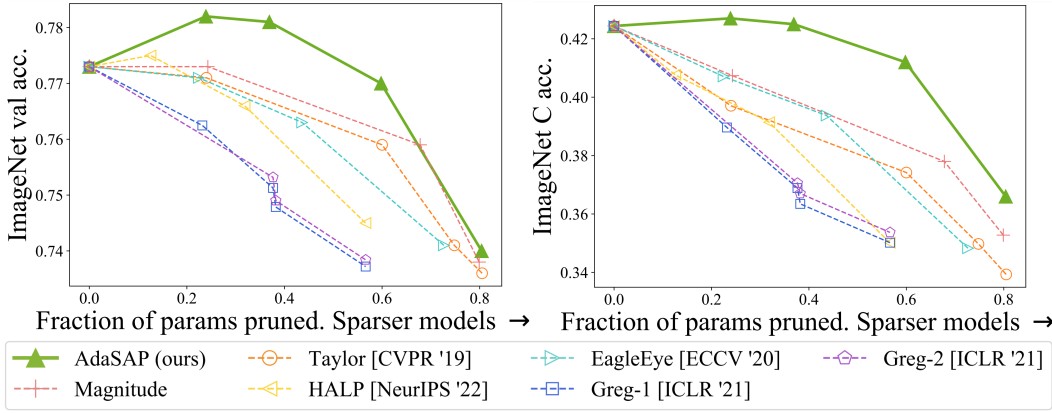

Figure 6: **ImageNet Validation and ImageNet C performance on AdaSAP vs. baselines.** Contains the same data as used to produce Figure 1 but demonstrates that AdaSAP additionally dominates baselines on both components of the relative robustness metric.

## C.2 VARYING THE LENGTH OF THE ADAPTIVE WEIGHT PERTURBATION STEP.

Throughout the main set of experiments use 10 epochs for the adaptive weight perturbation step. This delineates how long we use the AdaSAP optimizer for, as well as how soon into the procedure

Table 12: **Sensitivity to pruning criteria.** AdaSAP performs best when combined with magnitude pruning, but is flexible enough to be used with other criteria, such as Taylor importance. Here we see that AdaSAP with Taylor pruning matches or outperforms SGD with Taylor pruning. Results are on ResNet50 and do not use ASAM.

| Method | Size ↓ | Val | $R_{V2}$ | $R_C$ | IN-V2 | IN-C |
|---|---|---|---|---|---|---|
| Taylor + SGD | 0.42 | 75.85 | 0.84 | 0.50 | 63.51 | 37.84 |
| AdaSAP$_{P,\text{Taylor}}$ | 0.43 | 76.26 | 0.84 | 0.50 | 63.77 | 38.07 |
| **AdaSAP$_P$** | **0.41** | **76.93** | **0.84** | **0.52** | **64.49** | **39.64** |
| Taylor + SGD | 0.76 | 77.05 | 0.84 | 0.52 | 64.53 | 39.68 |
| AdaSAP$_{P,\text{Taylor}}$ | 0.76 | 77.42 | 0.84 | 0.52 | 65.24 | 40.27 |
| **AdaSAP$_P$** | **0.76** | **77.86** | **0.85** | **0.53** | **66.00** | **41.30** |

Table 13: **Confidence intervals over three repeats.** EagleEye checkpoints are obtained from the official repository with only one seed. Results are on ResNet50.

| Method | Size ↓ | Val | IN-V2 | IN-C |
|---|---|---|---|---|
| Magnitude | 0.20 | $73.71 \pm 0.12$ | $61.21 \pm 0.17$ | $35.33 \pm 0.18$ |
| Taylor | 0.20 | $73.42 \pm 0.19$ | $60.36 \pm 0.19$ | $33.97 \pm 0.12$ |
| EagleEye | 0.27 | 74.13 | 61.30 | 34.84 |
| **AdaSAP$_P$** | 0.20 | $\mathbf{74.54 \pm 0.09}$ | $\mathbf{62.21 \pm 0.13}$ | $\mathbf{37.30 \pm 0.19}$ |
| Magnitude | 0.76 | $77.32 \pm 0.06$ | $65.18 \pm 0.27$ | $40.64 \pm 0.20$ |
| Taylor | 0.76 | 77.05 | 64.53 | 39.68 |
| EagleEye | 0.78 | 77.07 | 64.84 | 40.67 |
| **AdaSAP$_P$** | 0.77 | $\mathbf{78.23 \pm 0.06}$ | $\mathbf{65.98 \pm 0.32}$ | $\mathbf{43.43 \pm 0.20}$ |

we begin pruning. In this experiment, we analyze the sensitivity of the approach to this parameter. We report results for this experiment in Table 15. We can observe that as we make this period longer, validation accuracy and ImageNet C accuracy both drop slightly, while ImageNet V2 seems to have no discernible pattern. Ratios $R_C$ and $R_{V2}$ also stay consistent across the experiment.

## C.3 ROBUSTNESS ENCOURAGEMENT

As mentioned in the main text, we consider various ablations to determine the necessity of various steps of the AdaSAP procedure. In the third step of our procedure, robustness encouragement, we choose to apply uniform perturbations across all weights in the network. This differs from the first step, in which we apply adaptive weight perturbations. In Table 16, we examine the effects of different weight perturbation strategies during the robustness encouragement phase. We can see that while all three strategies lead to relatively close final performance across the three datasets, uniform weight perturbations perform slightly better, suggesting that our choice of applying them in our procedure may be slightly benefitting the performance.

## C.4 IMPORTANCE OF ADAPTIVE WEIGHT PERTURBATIONS

In Table 17 we extend an ablation from the main paper in which we compare AdaSAP to SAM, effectively evaluating the importance of warmup with adaptive weight perturbations. Here, we perform the comparison on a wider range of sparsities and observe that a similar pattern emerges.

Table 14: Validation Loss and Accuracy directly before and after pruning (before additional fine-tuning). Across several pruning levels, our method generally reaches the lowest validation loss and accuracy both before and after pruning. Results are on ResNet50 and do not use ASAM.

| Method | Size ↓ | Val Loss Before | Val Loss After | Val Acc Before | Val Acc After |
|---|---|---|---|---|---|
| Taylor | 0.20 | **2.245** | 3.315 | **67.798** | 42.915 |
| Mag | 0.20 | 2.416 | 3.288 | 63.843 | 43.21 |
| SAM | 0.20 | 2.255 | 3.2 | 67.577 | 45.802 |
| AdaSAP$_P$ | 0.20 | 2.293 | **3.146** | 66.555 | **46.313** |
| Taylor | 0.43 | 2.284 | 2.69 | 66.67 | 57.046 |
| Mag | 0.43 | 2.408 | 2.587 | 63.844 | 59.275 |
| SAM | 0.43 | 2.327 | 2.676 | 65.541 | 57.449 |
| AdaSAP$_P$ | 0.43 | **2.251** | **2.574** | **67.473** | **59.962** |
| Taylor | 0.76 | 2.441 | 2.42 | 63.389 | 63.711 |
| Mag | 0.76 | 2.292 | 2.406 | 66.714 | 64.087 |
| SAM | 0.76 | 2.401 | 2.379 | 64.019 | 64.589 |
| AdaSAP$_P$ | 0.76 | **2.213** | **2.347** | **68.463** | **65.393** |

Table 15: **Effects of varying number of epochs of adaptive weight perturbation.** Increasing the number of epochs leads to smaller models but slightly worse validation and ImageNet C performance. Results are on ResNet50.

| Num epochs | Size ↓ | Val | $R_{V2}$ | $R_C$ | IN-V2 | IN-C |
|---|---|---|---|---|---|---|
| 5 | 0.48 | 73.89 | 0.83 | 0.48 | 61.29 | 35.65 |
| 10 | 0.46 | 73.82 | 0.83 | 0.48 | 60.91 | 35.59 |
| 20 | 0.44 | 73.63 | 0.84 | 0.48 | 61.48 | 35.25 |
| 30 | 0.43 | 73.47 | 0.83 | 0.48 | 60.72 | 35.04 |

Table 16: Comparison of various weight perturbation strategies during robustness encouragement. Results are on ResNet50.

| Perturbation Type | Val | IN-C | IN-V2 |
|---|---|---|---|
| No weight perturbations | 73.98 | 35.84 | 62.00 |
| Adaptive weight perturbations | 74.10 | 35.72 | 61.94 |
| Uniform weight perturbations | **74.39** | **35.86** | **62.03** |

Table 17: **Comparison of AdaSAP to SAM optimizer.** AdaSAP outperforms SAM and SGD on standard validation performance and ImageNet-C performance, but slightly trails SAM on ImageNet-V2 when both methods are augmented with ASAM. Results are on ResNet50.

| Method | Size $\downarrow$ | Val | $R_{V2}$ | $R_C$ | IN-V2 | IN-C |
|---|---|---|---|---|---|---|
| **SGD** | | | | | | |
| Dense | 1 | 77.32 | 0.84 | 0.54 | 64.79 | 42.46 |
| Taylor + SGD | 0.20 | 73.56 | 0.82 | 0.46 | 60.56 | 33.93 |
| **AdaSAP vs. SAM (without ASAM)** | | | | | | |
| Taylor + SAM | 0.20 | 73.62 | 0.83 | 0.47 | 61.37 | 34.49 |
| **AdaSAP$_P$** | 0.20 | **74.38** | **0.83** | **0.48** | **62.03** | **35.86** |
| Taylor + SGD | 0.42 | 75.85 | 0.84 | 0.50 | 63.51 | 37.84 |
| Taylor + SAM | 0.43 | 76.27 | 0.84 | 0.50 | 63.81 | 37.72 |
| **AdaSAP$_P$** | **0.40** | **77.03** | **0.84** | **0.51** | **64.62** | **39.57** |
| Taylor + SGD | 0.76 | 77.05 | 0.84 | 0.52 | 64.53 | 39.68 |
| Taylor + SAM | 0.76 | 77.29 | 0.84 | 0.52 | 64.84 | 40.07 |
| **AdaSAP$_P$** | 0.76 | **77.86** | **0.85** | **0.53** | **66.00** | **41.30** |
| **AdaSAP vs. SAM (with ASAM)** | | | | | | |
| SAM + ASAM | 0.19 | 73.93 | 0.84 | 0.50 | 61.76 | 36.66 |
| **AdaSAP$_P$ + ASAM** | 0.19 | **74.63** | 0.83 | 0.50 | **62.08** | **37.30** |
| SAM + ASAM | 0.41 | 77.10 | 0.84 | 0.53 | 64.94 | 40.90 |
| **AdaSAP$_P$ + ASAM** | **0.40** | **77.27** | 0.83 | 0.53 | 64.51 | **41.23** |

