# OpenReview forum: "Adaptive Sharpness-Aware Pruning for Robust Sparse Networks"
_ICLR.cc/2024/Conference — ICLR 2024 poster_

### Official Review · Reviewer_e15Q · 2023-10-28

**Soundness:** 3 good
**Presentation:** 3 good
**Contribution:** 2 fair
**Rating:** 6
**Confidence:** 4

**Summary:**

This paper proposes an adaptive sharpness-aware pruning including pre-pruning optimization and pruning procedures. It claims that the proposed method improves the robustness and efficiency and providing evaluation on different version of ImageNet.

**Strengths:**

* The evaluation of the proposed method is solid and comprehensive including robustness evaluation and the efficiency evaluation with different metrics.
* The presentation of the proposed method is clear and the paper is overall well-written.

**Weaknesses:**

* As mentioned by the author, SAM as a post-pruning training can help pruning has already been studied by several works, this work seems to be a modification or extension of using sharpness-aware neuron-level perturbation to enhance the pruning. So there exists a lack of novelty and clarification on the difference between this method and the vanilla SAM method, as well as the motivation for proposing AdaSAP optimization to surpass common SAM.
* There is no discussion of the hyperparameter tuning used in the method since it is especially critical for SAM-type methods.
* (minor) The paper may benefit from discussing a few more recent works [1][2] about the relationship between pre-pruning optimization and pruning performance in Related Works.

[1] [Peste et al.](https://arxiv.org/abs/2207.14200)
[2] [Zhou et al.](https://arxiv.org/abs/2305.18383)

**Questions:**

* In Section 4.5 and Table 4, will the results be consistent if we use some other way to measure the sharpness like Hessian Top-eigenvalue or trace?

**Details Of Ethics Concerns:**

No ethics concerns.

---

> ### Author Response · Authors · 2023-11-21
> **Response to Reviewer e15Q**
>
> Thank you for your thoughtful comments. We hope that we have addressed your points satisfactorily below.
>
> ### Weaknesses
> 1. In response to the first weakness, we agree that there are similarities between our method and SAM, and that there has been much work studying SAM in various settings. SAM was introduced as a method to improve (in-distribution) validation performance for (dense) models, while our goal is to improve OOD robustness of pruned models. We both study SAM in this novel context and introduce a new method that outperforms vanilla SAM. While we found that SAM can improve the robustness of pruned models (see Table 6 and Appendix Table 17), it was not intended to be used for pruning so it should be unsurprising that we can build on top of it to produce a better sharpness-aware pruning paradigm. The key difference in our method is that we introduce a novel adaptive weight perturbation, designed to be compatible with network pruning, which helps not only with standard validation performance, but also improves robust performance. Our adaptive perturbations penalize neurons that will be pruned differently than neurons which will be preserved post pruning, thus allowing the network to be optimally primed for pruning.
>
> 2. With regards to the second point, as described in appendix A.5, we choose our maximum $\rho$ based on the recommended values for $\rho$ in line with SAM and ASAM (Foret et al. 2020; Kwon et al., 2021). In our earliest experiments, we ran simple experiments where we evaluated the validation performance after one or two epochs (of training; no pruning involved) for varying $\rho_{min}$ and $\rho_{max}$. We tried minimum values of 0.01 and 0.05 and maximum values of 0.05, 0.1 and 0.5 (this formulation only considered perturbation balls, not other transformed shapes, and so we considered values close to the recommended $\rho=0.05$ in the SAM paper). We found here that the best $\rho_{min}$ and $\rho_{max}$ values were 0.01 and 0.05, respectively. When we added in the perturbation region transformations from ASAM, we only updated our maximum $\rho$ value to 2.0, in line with the recommended $\rho$ hyperparameter in the ASAM paper. We have added the following table to section A.5 of our paper (Table 7 in the appendix).
>
> | $\rho_{min}$ | $\rho_{max}$ | Epoch 2 Val Loss | Epoch 2 Val Acc |
> | ---|---|---|---|
> | $0.01$ | $0.05$ | $5.54$ | $8.63$ |
> | $0.01$ | $0.1$ | $5.59$ | $7.11$ |
> | $0.05$ | $0.1$ | $5.73$ | $5.69$ |
> | $0.05$ | $0.5$ | $6.92$ | $0.1$ |
>
> Remaining hyperparameters (i.e. learning rate, learning rate schedule, weight decay, warmup period, etc.) were fixed based on a previous SOTA paradigm with released code (HALP).
>
> 3. Finally, thank you for pointing out the two related works. We have cited them in our revised paper. Zhou et al. similarly analyzes SAM’s effectiveness as a pre-pruning optimizer, allowing for improved compression in certain regimes. Our work could perhaps be further refined by using the methods in this paper to vary the range of ($\rho_{min}$, $\rho_{max}$) based on the density of the model. As well, Peste et al. employs a similar approach to our paper - they modify the SAM objective to be more amenable to pruning and identify which weights would be preserved or pruned and optimize accordingly. CrAM primarily focuses on unstructured sparsity and does not evaluate OOD robustness, so the results are not immediately comparable to ours. However, it would be interesting to investigate under which conditions AdaSAP and CrAM perform well: one-shot vs gradual pruning, structured vs. unstructured sparsity patterns, standard validation vs OOD robustness performance.
>
> We have cited these two works near the end of page 3:
> > Some work builds upon SAM,
> including improving the efficiency of the method (Liu et al., 2022; Du et al., 2022), modifying
> the optimization for improved performance (Kwon et al., 2021), applying it for improved model
> compression in NLP applications (Na et al., 2022) and as a pre-pruning optimizer (Zhou et al., 2023),
> and modifying the objective to help with unstructured pruning (Peste et al., 2022).
>
> ### Questions
> 1. Yes, we would generally expect these results to be invariant to sharpness measure. We have some limited results measuring the sharpness via the top eigenvalue of the Hessian. We compare AdaSAP$_L$ to SGD with HALP pruning in the following table and observe that AdaSAP results in lower sharpness directly before pruning, directly after pruning, and after finetuning. We have added these results to Section B.5 and Table 11 in the Appendix.
>
> | Optimizer | Pre-pruning | Post-pruning | Post-finetuning |
> |---|---|---|---|
> | SGD | $35.91$ | $93.82$ | $8.17$ |
> | AdaSAP$_L$ | $25.23$ | $16.79$ | $7.66$ |

---

### Official Review · Reviewer_76yJ · 2023-10-30

**Soundness:** 3 good
**Presentation:** 2 fair
**Contribution:** 2 fair
**Rating:** 6
**Confidence:** 2

**Summary:**

This paper proposes to use a different optimization paradigm, that encourages flatter optima, for obtaining robust compressed models. The idea is based on the hypothesis that flatter minima encourage deep networks to be both robust and good generalisers. The optimization paradigm can be used with any exsiting pruning algorithm (demonstrated in experiments by using HALP and L2 norm structured pruning). The method is shown to improve both benign and adversarial performance on unseen OOD samples.

**Strengths:**

S1. The problem is well motivated and the justifications for the used solution makes sense.

S2. The experiments show marginal improvement in adversarial robustness upon using AdaSAP.

**Weaknesses:**

W1. Writing needs to be improved especially within section 3.1 where mathematical preliminaries need to be set well. Also, variables are being used much before their description/introduction.

W2. Lack of ablation with respect to $\rho_{min}, \rho_{max}$.

W3. Lack of adherence to ICLR template. Text is wrapped around images, tables, and algorithms.

**Questions:**

Q1. In each of the cases of AdaSAP pruning procedure, can you please report the accuracy numbers for the models post Adaptive Weight Perturbation Step (warmup)? It would be interesting to note the difference in numbers pre and post pruning.

Q2. When computing $\alpha_i$ (equation 2), did you intend to write $\rho_{max}$ instead on $s_{max}$ in the first term? The expression looks odd otherwise. If yes, can you explain the choice of using $\rho_i = - \alpha_i s_i$ ? It is unclear why would you require a negative sign. If no, can you please explain the choice behind defining $\alpha_i$ in such a way?

---

> ### Author Response · Authors · 2023-11-19
> **Response to Reviewer 76yJ**
>
> Thank you for your thoughtful comments.
>
> ### Questions
>
> Q1: We think your first question may be answered by Table 14 in the updated Appendix. This table shows the loss and accuracy right before (i.e. post Adaptive Weight Perturbation Step / warmup) and right after pruning. Our AdaSAP method generally reaches the lowest loss and highest accuracy both right before and right after pruning. This indicates that the model is primed well for pruning, and that the pruning procedure minimally impacts the performance. Please let us know if we can clarify these results further.
>
> (We also note that there was an omission in the initial version of this table which has since been updated - each grouped set of rows corresponds to a different pruning ratio. The first set of rows correspond to models with 20% of parameters preserved, or 80% pruned away; the second set is models with 43% of parameters preserved; and the third set is models with 76% of parameters preserved)
>
> Q2: We thank you for pointing out the typo here. You are correct in your analysis of Equation 2 - terminology was accidentally left from an alternative characterization. Equation 2 should be $\rho_i = \rho_{max} - \frac{s_i - s_{min}}{s_{max} - s_{min}} (\rho_{max} - \rho_{min})$. We appreciate this feedback and have updated the equation (as well as the reference in Algorithm 1).
>
> ### Weaknesses
> W1: We would love to improve the presentation of our paper based on your recommendations. We read carefully through the paper and found one instance in Equation 1 in which epsilon was only implicitly defined; we will change the feasible set in this equation to explicitly define epsilon: the maximization will be over $\epsilon:(\|T_{\mathbf{w}}^{-1}\epsilon_i\| \leq \mathbf{\rho_i})_{i=1}^I$. However, we could not find any more examples of variables being used much before their description. We would appreciate if you could provide any examples where this occurs, or any other specific writing that lacks clarity. If you’re referring to the places where we define terms in Equations 1 and 2 after we state the equations, please let us know if defining those terms before the equations would improve readability.
>
> W2: On $\rho_{min}$ and $\rho_{max}$, as described in Appendix A.5, we choose our maximum $\rho$ based on the recommended values for $\rho$ in line with SAM and ASAM (Foret et al. 2020; Kwon et al., 2021). In our earliest experiments, we ran simple experiments where we evaluated the validation performance after one or two epochs (of training; no pruning involved) for varying $\rho_{min}$ and $\rho_{max}$. We tried minimum values of 0.01 and 0.05 and maximum values of 0.05, 0.1, and 0.5 (this formulation only considered perturbation balls, not other transformed shapes, and so we considered values close to the recommended $\rho=0.05$ in the SAM paper). We found here that the best $\rho_{min}$ and $\rho_{max}$ values were 0.01 and 0.05, respectively. When we added in the perturbation region transformations from ASAM, we only updated our maximum $\rho$ value to 2.0, in line with the recommended $\rho$ hyperparameter in the ASAM paper. While we did not conduct an in-depth ablation of these hyperparameters, we found that extensive hyperparameter tuning was not needed to obtain our results and we simply chose consistent values that worked well from the beginning. We hope this can help clarify your question and have added this information to Appendix A.5 and Table 7 of the paper.
>
> | $\rho_{min}$ | $\rho_{max}$ | Epoch 2 Val Loss | Epoch 2 Val Acc |
> | ---|---|---|---|
> | $0.01$ | $0.05$ | $5.54$ | $8.63$ |
> | $0.01$ | $0.1$ | $5.59$ | $7.11$ |
> | $0.05$ | $0.1$ | $5.73$ | $5.69$ |
> | $0.05$ | $0.5$ | $6.92$ | $0.1$ |
>
> W3: With regards to your last point, we have updated the final paper to avoid text wrapping around the main tables. To the best of our knowledge, ICLR guidelines say nothing about text wrapping for figures and algorithms, but can further update the formatting as needed or to aid in clarity.
>
> We hope our responses above can help clarify the points you made and we would appreciate if there is any additional information we can add for you to consider revising your rating.

---

> > ### Comment · Reviewer_76yJ · 2023-11-23
> > **Thank you for your rebuttal.**
> >
> > The authors have more or less addressed my concerns. I am happy to increase my score from 5 to 6. However, I will request the authors to mention information regarding the model and dataset used in tables reporting experimental results. Such as in table 14 where it is not clear what was the setup (model, architecture) used for the experiments carried out.
> >
> > Particularly, for Table 14, can the authors report the same observations for other setups? Also, it would be convenient to see (loss/accuracy) numbers post additional finetuning in the same table as well.

---

### Official Review · Reviewer_xHva · 2023-11-01

**Soundness:** 3 good
**Presentation:** 3 good
**Contribution:** 3 good
**Rating:** 6
**Confidence:** 4

**Summary:**

The paper proposes AdaSAP, an optimization paradigm to produce sparse yet robust models against distribution shift. It unifies the goals of sparsity and robustness through the lens of loss landscape sharpness. The method has three steps: 1) Adaptive weight perturbations before pruning to push unimportant neurons into flatter minima, reducing the impact of pruning them; 2) Pruning unimportant neurons based on any criteria; 3) Sharpness-based optimization after pruning for overall robustness. Experiments on image classification and object detection show AdaSAP outperforms SOTA pruning techniques with higher robustness while maintaining accuracy. Analysis indicates AdaSAP results in flatter loss surfaces, explaining its robustness advantages. By effectively unifying sparsity and robustness through sharpness, this exploration presents a promising direction for robust and efficient models.

**Strengths:**

1. Unifying sparsity and robustness through the perspective of sharpness is an interesting idea.
2. Good results are achieved on both image classification and object detection tasks.

**Weaknesses:**

1. Although the idea to improve both sparsity and robustness sounds good, the method to improve robustness via weight perturbation is indeed not novel. Is there any special designs closely related to pruning?
2. In the experiments, the authors only conduct experiments in terms of robustness on convolutional networks. Nevertheless, there are a lot of works that have reported competitive robustness results, such as [a, b]. It is better to discuss or include these SOTA results into comparisons.
[a] "Understanding the robustness in vision transformers." ICML 2022.
[b] "Robustifying token attention for vision transformers." ICCV 2023.

3. While the proposed AdaSAP method demonstrates notable effectiveness in concurrently optimizing sparsity and robustness, it is better for the authors to include a discussion of other relevant works that aim to achieve such dual objectives. This would provide readers with a better understanding of where AdaSAP stands in this field and allow for comparisons with existing literature.
4. In the experiments on image classification, evaluating AdaSAP on additional pruning criterias would provide a more comprehensive assessment the effectiveness and generality of AdaSAP.
5. The experiments and analysis on object detection are inadequate with only one model. More detection models should be evaluated. Furthermore, comparisons with other pruning methods are needed to further verify efficacy.

**Questions:**

Please refer to the weakness part.

---

> ### Author Response · Authors · 2023-11-19
> **Response to Reviewer xHva**
>
> Thank you for your thoughtful comments.
>
> 1. We would like to highlight that the adaptive nature of our approach is what makes it particularly suited to pruning. With the knowledge that some of the neurons will eventually be pruned, we regularize the neurons with the lowest importance (i.e. penalize sharpness) so that they have a smaller impact on the loss when they are pruned. Without this adaptivity, our method reduces to Sharpness-Aware Minimization (Foret et al. 2020) and has no particular compatibility with pruning algorithms.
>
> 2. We thank you for pointing out those works. Neither of these papers focus on pruned models, and while they do improve the robustness of dense models, the objective of our work is to specifically improve the robustness of pruned models. Our aim is to address the observed (see Liebenwein et al., 2021; Hooker et al., 2019) degradation of robustness during model pruning and reduce this degradation. While we already had cited “Understanding the robustness in vision transformers" in our paper, we have now also added "Robustifying token attention for vision transformers" and clarified that these works pursue robustness in dense models, unlike our work which focuses on pruned models. We have highlighted the difference between these two papers in the last sentence on page 1:
> > Despite significant progress in this area, most existing robustness work focuses on dense networks (Guo
> et al., 2023; Zhou et al., 2022).
>
> 3. We believe that this work is the first to introduce a method which has the primary aim of improving the OOD robustness of pruned models. Robust pruning methods (discussed in Section 3.4) are the closest to our approach, but differ since these methods requires train-time access to the types of data corruptions observed at test time (which are often adversarial attacks in this setting), which we do not want to assume (Sehwag et al., 2020; Vemparala et al., 2021; Zhao & Wressnegger, 2022). Some additional works (Section 2, paragraph “Robustness”) have observed, but not attempted to address, reduced OOD robustness in pruned models (Liebenwein et al., 2021; Hooker et al., 2019) or analyzed adversarial robustness (Sehwag et al., 2020; Fu et al., 2021). This gap in the literature is what motivated the formulation of our method. If you have other relevant comparison points that we have missed, we would be happy to provide a more detailed analysis and include the references in our paper.
>
> 4. We have some additional results on applying AdaSAP to other pruning criteria. In Table 5 in the paper (and an expanded set of results in Table 12 in the appendix) we use AdaSAP with the Taylor pruning criteria and show that it improves performance.
> 5.
> We agree that the object detection results are limited. We will attempt to run additional baselines before the end of the rebuttal period, but we may be unable to obtain results in time.

---

### Official Review · Reviewer_Gd1H · 2023-11-03

**Soundness:** 3 good
**Presentation:** 2 fair
**Contribution:** 3 good
**Rating:** 6
**Confidence:** 4

**Summary:**

This paper presents a sharpness-aware structured pruning paradigm, AdaSAP, which can make a model more ready for pruning. The resulting model is more robust than other (sharpness-unaware) pruning methods. The proposed method has 3 steps, (1) Apply a proposed adaptive weight perturbations to optimization to adaptively penalize sharpness in order to prepare the network for pruning. (2) Prune the network (3) Continue training the model while uniformly penalizing sharpness across the network to encourage robustness. The key to improving robustness is step (1). Many specific pruning algorithms can be used in step (2). Empirically, the model of the proposed method is trained on the clean dataset (e.g., ImageNet), while showing superior performance than other SOTA structured pruning methods on OOD datasets (e.g., IageNet-C), in classification and detection tasks, with four networks.

**Strengths:**

1. Sharpness-aware pruning is an interesting topic and of rising importance these days since model robustness is attracting more attention.

2. The proposed method via weight perturbation is technically sound and sounds novel to me.

3. Empirically, the proposed method achieves superior performance than the other sharpness-unaware counterparts, showing the encouraging potential of the method.

**Weaknesses:**

1. Some of the experimental results look unconvincing. E.g., Taylor / GReg / ABCPruner are more advanced pruning methods than magnitude pruning, esp. for Taylor and GReg, their papers have shown they perform better than magnitude pruning. But here, in Tab. 1, for different sizes, Magnitude consistently performs better than the above three methods. I am wondering if the authors have correctly done their experiments.

2. Presentation.

2.1 Page 8, Tab. 1 and 2 seem too big and the text looks too narrow.

2.2 Typos and grammar mistakes.
- Tab. 1 caption: various -> various
- HALP (NeurIPS’23 (Shen et al., 2022b)) -> NeurIPS'22
- Sec. 4.5, first sentence misses a period.
-- These small glitches, not very serious though, make me feel this paper is sort of rushed out.

3. As mentioned in Sec. 4.6, the method requires twice the training time. The total training cost (e.g., how many GPU hrs) should be discussed in the paper.

**Questions:**

In Tab. 1, for the results of comparison methods, are they from the authors' rerun with the same base unpruned model, or, cited from the original papers?

---

> ### Author Response · Authors · 2023-11-21
> **Response to Reviewer Gd1H**
>
> Thank you for your thoughtful comments. Please let us know if any of the following points are still unclear.
>
> ### Questions
> Addressing your question about our comparison points: we ran magnitude, Taylor pruning, and HALP following hyperparameter specifications from the corresponding papers in the latter two. For all other comparison methods, we took numbers from their papers. We have clarified this in the paper (see the quote from Section 4.1 "Baselines" below).
>
> ### Weaknesses
>
> 1. We understand that the magnitude pruning results are unintuitive, however we believe that our experimental results are correct and consistent. The main point we want to emphasize is that AdaSAP, in combination with a particular pruning metric, should primarily be compared against a model trained with SGD (or another basic optimizer) and the same pruning metric. Therefore, magnitude is primarily intended as a comparison point with AdaSAP$_P$. We have attempted to clarify our results in the final paper in line with the following points.
> First, some of the discrepancy may come from variation in training procedures. This is unavoidable due to different setups across previous works. We used comparable training procedures when we ran both AdaSAP and magnitude pruning for our experiments, since AdaSAP utilizes L2-norm based magnitude pruning and we wanted to provide a direct comparison of our method to its closest baseline. Our goal in comparison to all other methods is primarily to demonstrate their lack of relative robustness out of the box. We would like to re-emphasize that many of these methods may be compatible with AdaSAP, which in turn could boost their performance. We demonstrate that AdaSAP + Taylor pruning outperforms Taylor pruning alone, and that AdaSAP + HALP outperforms HALP alone.
> Second, magnitude pruning is a strong baseline and results are often relatively close to SOTA methods. Therefore it should be unsurprising that small variations in training setups may lead to the type of results we observe.
> Finally, as a minor point, the comparisons to magnitude pruning in the original papers were slightly different: Greg compares to a L1-based magnitude method, whereas we use L2 and ABCPruner does not directly compare to magnitude pruning.
> We thank you for pointing out this discrepancy and we have clarified in the paper that magnitude pruning was intended primarily as a direct comparison to AdaSAP, since we did not rerun many other comparison methods from scratch.
>
> The updated text in the paper in Section 4.1 under "Baselines" now reads as follows:
> > We primarily focus on comparing to two SOTA baselines which we re-run in our experiments, Taylor pruning for parameter-based pruning (Molchanov et al., 2019), and HALP for
> latency-based pruning (Shen et al., 2022a). Taylor pruning assigns a score to each neuron based on a
> Taylor series approximation of how much the loss would change if the neuron were removed. HALP
> is a pruning method that optimizes for improving latency. We additionally run results on magnitude
> pruning (Han et al., 2015b), to demonstrate the improvements of AdaSAP$_P$ over its closest baseline.
> We also compare to the results cited in several other pruning methods such as GR-eg (Wang et al.,
> 2020b), EagleEye (Li et al., 2020), ABCPruner (Lin et al., 2020), and SMCP (Humble et al., 2022).
>
> 2. With regards to the misspellings, typos, and formatting, we appreciate your careful read and have fixed them in the final paper.
> 3. We agree that GPU hour statistics are beneficial to add. We were able to obtain some of these results and have augmented Section 4.6 with the following information:
> > One limitation is that AdaSAP requires twice the training time due to the two backward passes
> necessary for optimization. A 90 epoch pruning and finetuning run on 8 V100 GPUs to reduce a
> ResNet50 network to 0.2× its size takes takes 32 hours and 12 minutes for AdaSAP$_P$ while SGD
> with magnitude pruning takes 16 hours and 53 minutes. Recent strategies (Du et al., 2022; Liu et al., 2022) could be used to reduce this additional overhead.

---

> > ### Comment · Reviewer_Gd1H · 2023-12-04
> >
> > I checked the responses and other reviewers' comments. The responses have addressed my concerns. I'll keep my weak accept rating.

---

### Meta-Review · Area_Chair_hm6V · 2023-12-17

**Metareview:**

The paper introduces a method for producing sparse and robust (to input variation) networks by regarding both attributes as influenced by network sharpness.  After the author response and discussion, all reviewers favor accept, with Reviewers Gd1H, 76yJ, and e15Q indicating that the response addressed their concerns.  The AC agrees with the reviewer consensus.

**Justification For Why Not Higher Score:**

The paper adds to the existing body of work on pruning and robustness; overall contributions and reviewer evaluation are consistent with acceptance as a poster.

**Justification For Why Not Lower Score:**

The paper demonstrates solid experimental results using the proposed technique to improve robust accuracy of pruned models.

---

### Decision · Program_Chairs · 2024-01-16

Accept (poster)